# Examining the Effects of Mindfulness–Acceptance–Commitment Training on Self-Compassion and Grit among Elite Female Athletes

**DOI:** 10.3390/ijerph19010134

**Published:** 2021-12-23

**Authors:** Mahmoud Mohebi, Dena Sadeghi-Bahmani, Sahar Zarei, Hassan Gharayagh Zandi, Serge Brand

**Affiliations:** 1Department of Motor Behaviour and Sport Psychology, University of Tehran, Tehran 1439813117, Iran; mohebi.mahmoud@ut.ac.ir (M.M.); zarei.sahar@ut.ac.ir (S.Z.); 2Department of Psychology, Stanford University, Stanford, CA 94305, USA; bahmanid@stanford.edu; 3Center for Affective, Stress and Sleep Disorders, University of Basel, Psychiatric Clinics (UPK), 4052 Basel, Switzerland; 4Sleep Disturbances Research Center, Kermanshah University of Medical Sciences, Kermanshah 6719851451, Iran; 5Division of Sport Science and Psychosocial Health, Department of Sport, Exercise and Health, Faculty of Medicine, University of Basel, 4052 Basel, Switzerland; 6Substance Abuse Prevention Research Center, Kermanshah University of Medical Sciences, Kermanshah 6719851451, Iran; 7School of Medicine, Tehran University of Medical Sciences, Tehran 1417466191, Iran

**Keywords:** Mindfulness–Acceptance–Commitment intervention, active control intervention, females, elite athletes, grit, self-compassion

## Abstract

Background: Mindfulness-based interventions are well-established in the field of psychotherapy, and such interventions have also gained increased attention in the field of sport psychology, either to cope with psychological pressure or to improve an athlete’s performance. The goal of the present study was to examine whether a Mindfulness–Acceptance–Commitment (MAC) program could increase self-compassion and grit among elite female athletes compared to an active control condition. To this end, we performed a randomized trial among female adult athletes. Methods: Forty female adult athletes (M_age_ = 22.22, SD = 2.40) were randomly assigned either to the Mindfulness–Acceptance–Commitment group (*n* = 20; 7 group sessions, 60 min each) or the active control group (*n* = 20; 7 group sessions, 60 min each). At baseline, seven weeks later at the end of the study and again four weeks later at follow-up, participants completed a series of self-rating questionnaires on mindfulness, self-compassion and grit. Results: Dimensions of mindfulness, self-compassion and grit improved over time, but more so in the Mindfulness–Acceptance–Commitment condition compared to the active control condition. Improvements remained stable from the study end to follow-up. Conclusions: While the active control condition improved dimensions of mindfulness, self-compassion and grit among female adult athletes, improvements were much stronger in the Mindfulness–Acceptance–Commitment condition. Importantly, improvements in the Mindfulness–Acceptance–Commitment condition remained stable over a time lapse of four weeks at follow-up after study completion, suggesting that the Mindfulness–Acceptance–Commitment intervention appeared to improve cognitive–emotional learning processes.

## 1. Introduction

Among elite athletes, individual skill-based and physical differences are minimal [1] and psychological skills have become key factors for an athlete’s performance and success. Given this, such psychological factors came into the focus of attention of athletes, coaches and sport psychologists [1]. This holds particularly true, as there is increasing evidence of mental health issues among heavy exercisers [2,3,4]. Strikingly, 12–14% of promising junior elite athletes reported symptoms of burnout [5,6].

Regarding the optimization of psychological processes, the Optimal Competitive State Theory [7] also emphasized that optimal psychological status was the basis for athletes to maximize their performance under competition conditions.

To improve an athlete’s psychological state, relaxation techniques in general [8], specifically mindfulness, have been integrated into psychological skills training in the sport setting [9]. In general, relaxation techniques such as autogenic training (AT), imagination (IM) or progressive muscle relaxation (PMR) focus on the athlete’s personality development, including body awareness, mental calm, psychological wellbeing and avoidance of overtraining [8]. Complementarily, Gardner and Moore [10,11] introduced the Mindfulness-Acceptance–Commitment (MAC) intervention, which is the most common mindfulness approach in sport contexts specifically designed for athletes. Mindfulness is the state of “being attentive to and aware of the present moment” [12] and the state of “paying attention to experiences without passing judgment or fixating on one thing” [13].

Compared to the traditional psychological skill training (e.g., relaxation training and imagery training), which follows the rationale of eliminating or controlling the unpleasant psychological experiences to attain optimal psychological states [14], MAC training is an alternative approach for athletes to experience their psychological states. More specifically, according to the MAC approach, athletes’ optimal performance may be enhanced by a nonjudgmental, present-centered awareness of outer stimuli and inner experience, which helps emancipate distractions from their ruminative experience [10,11]. In other words, mindfulness training (e.g., MAC intervention) may help athletes to elude disadvantageous or ineffective psychological experiences [9]. To illustrate, 42 male semi-professional soccer players reported that MAC training could increase mindfulness and facilitate performance [15]. Specifically, Zadeh, Ajilchi et al. [15] concluded that the intervention resulted in greater improvements in post-test scores of mindfulness, which remained constant in long-term follow-up. Incidences of injuries were significantly lower, while individual and team performance was significantly greater, following MAC. Mindfulness scores were correlated at all time points. Higher scores of mindfulness at post-test and follow-up were associated with lower incidence rates of sport injuries. However, no effect sizes were reported in this study. Similarly, Zhang, Si et al. [16] reported that compared to a passive control condition, the mindfulness group had increased scores of mindfulness and dart throwing performance scores after the MAC training. Such improvements remained stable until follow-up. 

Further, interventions of Mindfulness Based Stress Reduction (MBSR) proved to be helpful in reducing symptoms of anxiety, depression and stress after the transition from professional football to retirement among a sample of Iranian male professional football players compared to an active control condition [17]. Specifically, Norouzi et al. [17] randomized forty male retired football players (mean age: 34 years) either to the MBSR condition or to the active control condition. The intervention lasted eight weeks, with a follow-up assessment again 12 weeks later. Compared to the active control condition, symptoms of depression, anxiety and stress decreased, and dimensions of wellbeing increased. Improvement remained stable from the end of the study to follow-up. Effect sizes were medium to large. The beauty of this study was the standardized implementation of an MBSR program, the introduction of an active control condition [18,19,20,21] and the report of effect sizes. Based on these results, the first aim of the present study was to investigate if a specific MAC intervention could improve dimensions of mindfulness and self-compassion among female adult athletes compared to an active control condition.

In the attempt to explain why some athletes outperform both at short- and long-term, besides the personality trait of resilience [1,22,23,24,25] and mental toughness [26,27,28,29,30,31,32,33], the concepts of grit and self-compassion (SC) have drawn attention in the field of sport psychology and elite athlete performance. This holds particularly true, as both grit and self-compassion are understood as favorable adaptation processes in the face of adversity, difficulty or stress in sport [34,35]. Self-compassion is a positive way of dealing with oneself when confronted with personal difficulties and challenging situations in sport [34]. Grit is characterized by passion and perseverance for achieving a main goal (or goals) despite barriers during improvement [36]. Grit and SC are considered to be critical psychological states that athletes can acquire to be successful in sport settings [37,38]. Both grit and self-compassion are rooted in the concept of positive psychology [39]. Athletes with high grit and self-compassion overcome obstacles, perform optimally in high-pressure environments, pursue the level of excellence and enjoy long and successful careers [40,41]. In important competitive events, female athletes utilized self-compassion to promote performance perceptions and wellbeing when preparing, competing and reflecting to excel in sport [38]. Specifically, Barczak and Eklund [42] found that athletes’ evaluations of their perceived performance after a performance episode were different based on their self-compassion levels. Concerning grit outcomes, several studies have found positive associations between grit and performance in the different sport types [43,44,45]. For example, results of a longitudinal study showed that higher scores of grit predicted both high endurance and performance in professional footballers [46].

Mindfulness, SC and grit are interrelated, as recent studies explained [47] the benefits of mindfulness training on SC. Theoretically, mindfulness (e.g., the MAC approach) can help individuals to be open to and accepting of their inner experiences without judgment in the face of difficulty and unexpected stressors in performance-related situations [9]. Complementarily, self-compassion entails being kind and nonjudgmental towards oneself when faced with pain, inadequacy, suffering and failure [34]. With regard to the empirical evidence of the neural mechanisms of mindfulness, mindfulness training was associated with reduced activations in cortical regions associated with self-critical cognitions, which in turn may be mediated by increased self-compassion [48]. The findings suggested that mindfulness and self-compassion could act as buffers against the negative effects of self-criticism on performance and overall wellbeing [49,50]. In addition, mindfulness could be a mechanism through which SC reduces scores of burnout in female athletes [51] because being mindful is a component of SC [52].

When discussing athletes’ mindfulness and sport performance, these dimensions were associated with grit [53]. Grit is conceptualized as passion, endurance and tenacity [54] for long-term goals and has been associated with optimal performance [43]. Mindfulness-based approaches had the potential to instigate skills of positive attention, nonjudging mindful awareness and experiential acceptance to aid in the pursuit of valued goals [11]. Thus, through mindfulness, athletes were enabled to regulate their behaviors to align with their performance goals; that is to say, their level of grit improved. Previous investigations have shown that higher levels of mindfulness have been associated with greater grit in females athletes [55]. Specifically, when comparing a resilience training program including Cognitive Behavioral Therapy (CBT) with mindfulness and positive psychology for college student athletes, the majority of participants indicated that the most important component from the program was mindfulness/meditation to develop grit [56]. In contrast, a brief mindful yoga intervention did not enhance grit among male athletes [57]. However, the limitations of the above mentioned studies were the lack of randomization and differences in the recruitment procedure, the lack of follow-up measures and the lack of control conditions.

Female athletes were dealing with many potentially challenging events in competitive sport, and such challenging events were often accompanied by negative psychological emotions (i.e., feelings of inadequacy, negative self-judgment and emotional disruption) and poor performance [41,58]. To counter such dysfunctional emotions, grit and self-compassion had the potential to improve psychological states in female athletes, along with their sport performance [37,38,59]. These findings appeared particularly important, as female athletes reported lower scores of mindfulness, grit and self-compassion, when compared to their male counterparts [51,60]. As such, focused efforts to understand effects of a mindfulness program on grit and self-compassion among female athletes appears plausible. Given this, the second aim of the present study was to investigate if a specific MAC intervention could improve grit among female adult athletes compared to an active control condition.

Gardner and Moore [10] created and developed the Mindfulness–Acceptance–Commitment (MAC) program; the MAC is a semi-structured fusion of mindfulness and Acceptance and Commitment Therapy (ACT). Mindfulness- and acceptance-based approaches develop skills and attitudes of nonjudging mindful awareness, mindful attention and experiential acceptance to aid in the pursuit of valued goals [14]. The MAC approach was designed and developed specifically for sport performers to develop mindfulness and self-regulated attention skills [11]. Gardner and Moore [10] claimed that dysfunctional and rigid thinking about one’s abilities or potential may create an avoidant coping style, and the individual will be more likely to disengage from the activity at hand. Essentially, mindfulness practices can break such a dysfunctional pattern. With open-hearted acceptance of what is occurring when faced with aversive experience, such as self-doubt and fear, athletes can disengage from negative thinking (that is to say, change their relationships to it) and be freed up to attain optimal performance states, so that athletes are enabled to learn they can reach peak performances despite negative internal states and negative self-evaluations and expectations [14]. Enhanced performance has been reported after implementing the MAC protocol with different samples of athletes, including female power lifters [10], professional lacrosse players [11], semiprofessional football players [15] and elite triathletes [61].

To summarize, compared to male athletes, female athletes appeared to be at an increased risk to report negative psychological emotions (i.e., feelings of inadequacy, negative self-judgment and emotional disruption) and poorer performance [62,63,64,65]. Mindfulness-based interventions in general, and MAC, specifically, proved to be useful to counterbalance dysfunctional emotional states [1,66,67,68] and to improve psychological wellbeing [50,69]. In the present study, we investigated the influence of a MAC intervention to improve mindfulness, grit and self-compassion among female athletes compared to an active control condition.

The following two hypotheses were formulated. Based on previous research [50,66,68], we first expected that a Mindfulness–Acceptance–Commitment (MAC) intervention would increase the scores of the three outcome measures (mindfulness, grit and self-compassion) among female athletes compared to an active control condition. Second, following previous research [15,50], we expected that after a MAC intervention, improvements would remain stable from the end of the study to the follow-up assessment four weeks later.

## 2. Methods

### 2.1. Study Procedure

Female athletes at national competition level were approached to participate in the present study. All eligible participants were informed about the aims of the study and the confidential data handling. Thereafter, they signed the written informed consent. Participants were randomly assigned either to the Mindfulness–Acceptance–Commitment condition or to the active control condition. At baseline, seven weeks later at the end of the study and again four weeks later at follow-up, participants completed a series of self-rating questionnaires (see below) covering dimensions of mindfulness, self-compassion and grit.

The ethical committee of University of Tehran (Tehran, Iran) approved the study (REC.1400.025), which was performed in accordance with the seventh and current [70] edition of the Declaration of Helsinki.

### 2.2. Participants

Participants were female athletes at national competition level. Inclusion criteria were: 1. Biological sex at birth: female; 2. age: ≥18 years; 3. Records of at least one success at national level or above; 4. Heavily exercising at least three times per week; 5. Compliance with the study conditions; 6. Signed written informed consent. Exclusion criteria were: 1. Current injuries or physical and mental health problems, as self-reported; 2. Previous or current experiences in relaxation and mindfulness techniques, including MBSR, progressive muscle relaxation, autogenic training and psychological skill training; 3. Missing two or more training sessions, irrespective of the study condition.

To recruit possible participants, six teams were selected randomly (*n* = 110 athletes; see Figure 1), and 42 female athletes were again randomly selected. Before randomization, two participants withdrew from the study for personal reasons, and thus 40 athletes were randomly assigned either to the MAC training condition or the active control condition (each group, *n* = 20) using a random number generator. There was no attrition in either the intervention or active control group after randomization (see Figure 1). Athletes were involved in various types of sport at the time of the study.

### 2.3. Sample Size Calculation

The sample size calculation was performed with G⃰ Power^®^ [71]. Based upon previous studies on the effects of mindfulness-based intervention on performance-relevant parameters and performance outcomes in sport [47,72,73], we expected a medium effect size of partial eta-squared 0.06; given this, the following sample size was calculated: partial eta-squared: 0.06; effect size f: 0.25; alpha: 0.05; power 1-beta: 0.80; number of groups: 2; number of measurements: 3; total sample size: 28. However, to counterbalance possible dropouts, the sample size was set at 42 participants.

### 2.4. Interventions

#### Mindfulness–Acceptance–Commitment (MAC) Program

In the span of seven consecutive weeks, participants in the intervention group underwent the Mindfulness–Acceptance–Commitment (MAC) program [11], guided by two trained sport psychologists with expertise in mindfulness. Each session covered specific topics and exercises within the context of mindfulness training and practice, which lasted 60 min/session (see Table 1). The task of the MAC instructors was to train a number of techniques and experiential exercises aimed at increasing mindful awareness and nonjudgmental acceptance of in-the-moment sensory, cognitive and affective experiences in the lives of the athletes. Further, MAC uses techniques to help athletes to develop the willingness to remain in contact with the full range of internal experiences in order to engage in behaviors consistent with their personal values. Additionally, in accordance with the guidelines of MAC protocol [11], the mindfulness group was asked to practice mindfulness at least three times per week. To ensure that athletes completed their practice at home, we asked about the recorded mindfulness exercises in the beginning of the training session.

### 2.5. Active Control Condition

The MAC program was organized as group intervention, and two trained sport psychologists with expertise in mindfulness were responsible for that program. Thus, there might have been the risk that improvements in outcome variables (mindfulness, self-compassion and grit) might have been due to the group setting and due to the contact with professional sport psychologists. To counterbalance this risk and thus to control for social interactions and social contacts as a possible confounder, participants of the active control condition met weekly for seven consecutive weeks and sessions (duration: 60 min per session). In groups, these participants attended a course on sport and exercise psychology. Topics included history, research, theories, talent identification and athletes’ lifespan development. Additionally, according to Zhang, Si et al. [16], participants of the active control group conditions were asked to revise the learning materials of the previous lecture at least three times per week. Two experienced sport psychologists gave these lectures; to enhance students’ participation, participants completed regular short and easy-to-complete exams. All sessions took place in the sport center of the University of Tehran (Tehran, Iran), and the intervention and active control conditions did not differ with regard to number, duration and format of the sessions or qualifications of the therapists.

### 2.6. Measures

#### 2.6.1. Sociodemographic and Sport-Related Information

Participants reported their biological sex at birth (female), age (years), civil status (single; married), the highest educational level (high school; bachelor; master), weekly sport frequency (trainings/week), duration (min/training), and intensity (light, moderate, vigorous) and the highest competition record within the last two years.

#### 2.6.2. Mindfulness

To assess mindfulness, participants completed the Farsi version [74] of the Mindfulness Inventory for Sport (MIS) [75]. The MIS consists of 15 items designed to determine athletes’ mindfulness levels. Dimensions include nonjudgmental thought, re-focusing and awareness. Typical items are: “When I become aware that I am not focusing on my own performance, I blame myself for being distracted.”, “When I become aware that I am not focusing on my own performance, I am able to quickly refocus my attention on things that help me to perform well” and “I am aware of the thoughts that are passing through my mind”. Answers are given on six-point rating scales ranging from 1 (=not at all) to 6 (=very much), with higher sum scores reflecting higher mindfulness. The Farsi version showed satisfactory psychometric properties [74], and the internal consistency of the current sample was 0.89 (Cronbach’s alpha at baseline).

#### 2.6.3. Self-Compassion

To assess self-compassion, participants completed the Farsi version [76] of the Short Self-Compassion Scale (SCS-SF) [77]. The SCS-SF includes dimensions of mindfulness, self-kindness and common humanity and consists of 12 items. Typical items are: “When something painful happens I try to take a balanced view of the situation”, “I’m disapproving and judgmental about my flaws and inadequacies” and “When I fail at something that’s important to me, I tend to feel alone in my failure”. Answers are given on five-point rating scales with the anchor points 1 (=almost never) to 5 (=almost always). After reverse-coding the appropriate items, higher scores reflect greater self-compassion. The Farsi SCS-SF showed good internal consistency [76], and the internal consistency of the total scale was satisfactory in the current sample (Cronbach’s alpha at baseline = 0.88).

#### 2.6.4. Grit

To assess grit, participants completed the Farsi version [37] of the Short Grit Scale (SG-S) [35]. The SG-S consists of eight items, and the measure includes dimensions of reflecting the tendency and perseverance of effort. Typical items are: “Setbacks don’t discourage me.”, “I have difficulty maintaining my focus on projects that take more than a few months to complete” and “I am a hard worker”. Answers are given on five-point Likert scales ranging from 1 (=very much like me) to 5 (=not much like me at all). After reverse-coding the appropriate items, higher scores reflect greater grit. The Farsi version showed satisfactory internal consistency [37], and Cronbach’s alpha of the total scale in the current sample was 0.78 at baseline.

### 2.7. Data Analysis

A series of independent sample *t*-tests and *X*^2^-tests was performed to compare sociodemographic information at baseline between participants in the MAC and in the active control condition. Next, three rANOVAs were performed to test the effect of group (MAC group vs. active control), time (baseline, post-intervention and follow- up) and the time by group interaction on mindfulness, self-compassion, and grit. The nominal significance level was set at alpha < 0.05. For F-tests, partial eta-squared (*η_p_*^2^) effect sizes are reported. Cut-off values for partial eta-squared were: *η_p_*^2^ < 0.019: trivial effect size (T); 0.02 < *η_p_*^2^ < 0.059: small effect size (S); 0.06 *< η_p_*^2^ < 0.139: medium effect size (M); *η_p_*^2^ > 0.14: large effect size (L) [78,79]. Based on Becker’s approach to comparing mean changes [80], Cohen’s d effect sizes were reported for the pre- and post-trial changes within the two groups and between the two groups at the end of the study. Effect sizes for t-tests were reported as Cohen’s d with the following ranges: d = 0 to 0.19: trivial effect sizes (T); d = 0.20 to 0.49: small effect sizes (S); d = 0.50 to 0.79: medium effect sizes (M); d ≥ 0.80: large effect sizes (L) [78,79]. All statistical calculations were performed in SPSS^®^ version 26.0 (IBM Corporation, Armonk, NY, USA) for Windows^®^.

## 3. Results

A total of 42 eligible female athletes participated in the study. Of the 42 athletes who participated before randomization, 40 completed all phases of the program. The average age and sport experience were 22.22 years (*SD* = 2.40 years) and 8.30 years (*SD* = 1.78 years), respectively. Furthermore, their average exercise was 3.80 days a week (*SD* = 0.82 days).

Table 2 provides participants’ sociodemographic and exercise-related descriptive and inferential statistical overview, separately for participants in the MAC and active control conditions. Participants in the MAC and active control conditions did not differ regarding age, sport experience, number of session training per week, intensity of training, marital status and highest educational level (*p* > 0.05).

### Mindfulness, Self-Compassion, Grit; between the Study Conditions and Over Time

Table 3 provides the descriptive statistical indices of MAC-related information, Table 4 reports the inferential statistical indices and Figure 2 reports the means and standard deviations of grit scores, separately for the MAC condition and for the active control condition.

Mindfulness increased over time (large effect size), but more so in the MAC condition (medium effect size) compared to the active control condition (large effect size of interaction).

Self-compassion increased over time (large effect size), but more so in the MAC condition compared to the active control condition (large effect size of interaction). Scores were higher in the MAC condition compared to the active control condition (medium effect size).

Grit also increased over time (large effect size), but more so in the MAC condition compared to the active control condition (large effect size of interaction). Scores were higher in the MAC condition compared to the active control condition (medium effect size).

Table 5 provides the overview of effect size calculations (Cohen’s ds) between the MAC and active control conditions at the end of the study and within the MAC and active control conditions from baseline to the end of the study.

At the end of the study and compared to the active control condition, MAC improved mindfulness (large effect sizes). Large effect sizes also were observed for SC and grit.

Within the MAC condition, large effect sizes were observed for mindfulness, SC and grit.

Within the active control condition, small effect sizes were observed for SC; trivial effect sizes were observed for mindfulness and grit.

## 4. Discussion

The key findings of the present study were that among young adult female athletes a seven-week Mindfulness–Acceptance–Commitment (MAC) intervention improved mindfulness, self-compassion (SC) and grit compared to an active control condition. Given this, the present data expanded upon the current literature in the following four ways. First, unlike previous studies, the introduction of an active control condition precluded the risk that improvements were the mere result of social interactions and group dynamics. Second, the MAC intervention also improved grit. Third, while improvements were also observed among participants in the active control condition, such improvements were modest. Fourth, among participants in the MAC condition, improvements remained stable over a time lapse of four weeks until follow-up, suggesting thus that the MAC program changed participants’ cognitive–emotional information processing.

In our opinion, the present results are of clinical and practical importance because interventions such as the MAC program appeared to enable (here: female) athletes to proactively and successfully cope with psychological pressure and to improve dimensions of grit as a proxy of resilience and mental toughness.

Two hypotheses were formulated and each of these is considered now in turn.

With the first hypothesis, we assumed that compared to an active control condition, a MAC intervention would improve the dimensions of mindfulness, grit and self-compassion among female athletes, and data fully support this. Accordingly, the present data confirm what has been observed elsewhere [47,50,66,68]. However, the present data expand upon previous research [47,50,66,68] in that a MAC intervention was applied among young adult female athletes and in that the intervention improved not only mindfulness and self-compassion but also grit.

With the second hypothesis, we assumed that after a MAC intervention, improvements would remain stable from the end of the study to the follow-up assessment four weeks later, and again, data confirm this. As such, the present results mirror what has been observed before [15,50]. However, the present findings expand upon previous research [15,50] in that a MAC intervention proved to be stable, not only for the dimensions of mindfulness and self-compassion but also grit, once the interventions were complete. The quality of the data does not allow a deeper understanding as to why improvements in the MAC condition remained stable over time. Admittedly speculatively, we claim that the MAC intervention over a time lapse of seven weeks led to a learning process for athletes on how to deal with their emotions and cognitions such that participants in the MAC condition were able to implicitly transfer and employ their acquired knowledge over a short time lapse of four weeks.

The quality of the data does not allow a deeper understanding of the underlying neuroendocrinological and cognitive–emotional processes underlying why the administration of Mindfulness–Acceptance–Commitment (MAC) protocol improved SC and grit. Thus, we can only speculate about which factors the observed changes in SC and grit can be attributed to. To gain a more solid understanding of the psychological mechanisms behind the observed effects associated with the MAC intervention, more research is warranted.

As described in Table 1, the aim of our MAC program was to perceive cognitions and emotions without reacting to them, to accept one’s personality traits, to remain positively disposed and to experience moments of time in peace. More specifically, as outlined elsewhere [50,66,68], MAC does not aim to change thought content, emotions or bodily sensations, but rather to focus on how one relates to such experiences. Self-compassion is also a positive way of relating to oneself, specifically when one is psychologically suffering in some way. Neff and Dahm [81] argued that in order to give oneself compassion, one must be able to turn toward, acknowledge and accept that one is suffering, meaning that mindfulness is a core component of self-compassion. Addressing compassion more broadly, Gilbert and Choden [82] viewed mindfulness as a context through which compassion is achieved, necessary for both the motivation to engage with suffering and the skill to alleviate it [83]. In a recent population-based study, Wilson, Bennett [52] further identified mindfulness as a critical key in order to develop and maintain self-compassion in competitive sport contexts. Therefore, it appears that authors view self-compassion as being developed implicitly through attitudes cultivated in mindfulness training. In line with these observations, researchers [10,11,84] argued that mindfulness-based programs (e.g., MAC) bring about mindfulness via an attitude of awareness, acceptance, kindness, openness, patience, nonstriving, equanimity, curiosity and nonevaluation.

To conclude, these findings suggest that the properties of MAC training set the stage for developing self-compassion by encouraging female athletes to be open and accepting of their inner experiences without judgment and interference in the face of sport-related difficulty and unexpected stressors.

Regarding grit, another finding of the present study indicated that increasing mindfulness leads to improved grit of female athletes, which is in line with the previous results [55]. On the other hand, the results of a further study indicated that brief mindfulness training could not help improve male athletes’ grit [57]. Two key limitations of this study design [57] were the lack of randomization and differences in recruitment. Further, it may be necessary to use a longer period of mindfulness training (e.g., MAC) to establish statistically significant changes in grit when assessed quantitatively.

As mentioned earlier, MAC enables athletes to increase their willingness to accept negative thoughts and emotions in the pursuit of valued ends [11]. More specifically, MAC interventions should enable athletes to mindfully engage in the present moment and help athletes to clarify and to formulate their valued goals and to engage in committed actions towards these goals [11]. Given this, admittedly speculatively, we claim that in the event of negative feelings, an athlete might answer in a gritty fashion; such an answer may be given to experience and tolerate difficult emotions and to remain motivated to pursue their goals. Thus, it appears conceivable that through a MAC approach, athletes were able to regulate their behaviors in accordance with their goals; in doing so, athletes might have improved their level of grit. In addition, it appears that mindfulness improves the interoceptive attention, which in turn is believed to increase cognitive flexibility and cognitive appraisal [85,86].

The novelty of the findings should be considered within the context of the following limitations. First, the present study relied on self-report measures, though, to our knowledge, the means to reliably measure cognitive–emotional processes of mindfulness or self-compassion are sparse [87]. Second, it is possible that the power of the findings could have been increased by including other indices of change for emotional, behavioral and sport performance outcomes. Third, the sample size might be considered small, though we mainly relied on effect size calculations, which by definition are not sensitive to sample sizes; further, the sample size matches the requirements of effect size calculations. In this view, we also note that effect sizes were medium to large, which were somehow higher when compared to previous studies. Fourth, we acknowledge that our sample was homogeneous regarding the gender and nationality of the participants. Thus, whether the conclusion of this study is applicable to male professional athletes or athletes from other countries needs further verification. Fifth, there were no objective measures of whether the intervention improved athletes’ objective performance because we deemed it impossible to develop a general objective performance measure that would be sufficiently adequate for a multiple-sport sample, consisting of both individual- and team-sport athletes. Sixth, while participants in the MAC condition were specifically trained on dimensions of mindfulness and self-compassion, participants in the active control condition were not. As such, it might come by no surprise that participants in the MAC condition showed higher scores in mindfulness and self-compassion. However, the key intention of the active control condition was to control outcome variables for possible biases related to the social setting of the group interaction and the interaction with professional sport psychologists. Seventh, participants were not blind to the treatment status; as such, it is conceivable that expectations might have biased the pattern of results. Last, to assess grit, we employed the Farsi version of the Short Grit Scale; the Farsi measure has not been psychometrically validated so far. Moreover, the concept of grit has faced several criticisms [53,88], and further research about this construct is warranted.

## 5. Conclusions

Overall, this study expands the functions of mindfulness-based training in the field of elite sport and focuses on the effects of a Mindfulness–Acceptance–Commitment (MAC) intervention on self-compassion and grit, which are closely related to female athletes’ performance and mental health. We highlight the importance of preserving the habit of regular mindfulness practice (e.g., MAC) as part of the process of female performance preparation in order to deal effectively with difficult sport experiences and to maximize performance.

## Figures and Tables

**Figure 1 ijerph-19-00134-f001:**
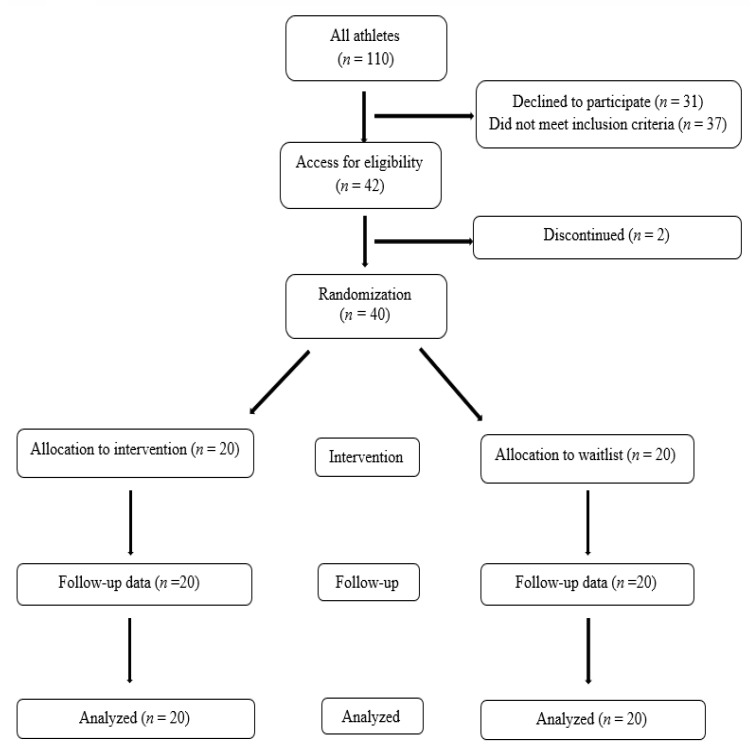
Consort flow diagram.

**Figure 2 ijerph-19-00134-f002:**
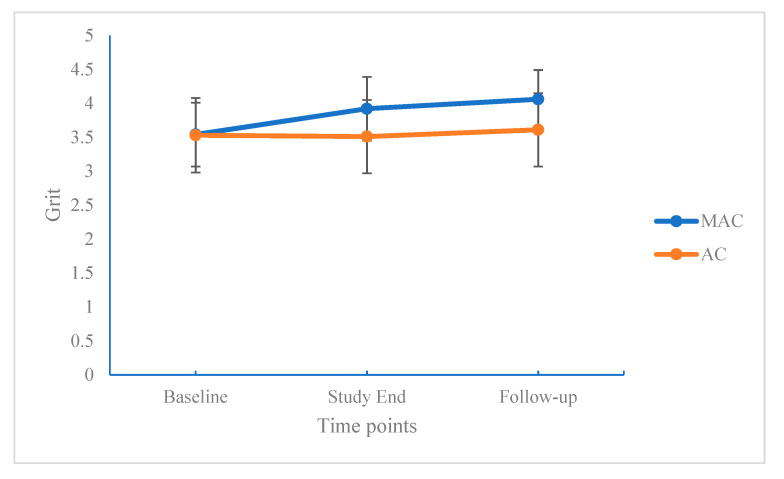
Grit increased over time, but more so in the MAC condition compared to the active control condition. Improvements at the end of the study remained stable until follow-up 4 weeks later. Points are means and bars are standard deviations.

**Table 1 ijerph-19-00134-t001:** Mindfulness–Acceptance–Commitment Training (MAC) Session Content [11].

Theme of Each Session	Contents
Session 1: Introduction	Introduction to the theoretical structure and content of the full MAC program; acknowledging negative thoughts/feelings Mindfulness exercises (e.g., mindfulness of breath)
Session 2: Mindfulness	Introducing mindfulness and cognitive defusion (vs. cognitive fusion) and how these concepts can be applied in a sport context Mindfulness exercises
Session 3: Goals and Values	Information about goals, values and behaviors, and the relation between them is presented and discussed The benefits of value-based behavior rather than emotion-focused behavior is emphasized Mindfulness exercises
Session 4: Acceptance	Developing an understanding of the costs of experiential avoidance and the benefits of experiential acceptance when striving for improved performance Mindfulness exercises
Session 5: Commitment	Introducing motivation and commitment and how they differ from one another Information about the relationship between motivation and commitment with performance-related values and behaviors is presented and discussed Mindfulness exercises
Session 6: Skill Consolidation	Integrating learned concepts to attain and maintain behavioral flexibility in the pursuit of long- and short-term goals Mindfulness exercises
Session 7: Skill Maintenance	Reviewing the entire MAC approach and encouraging athletes to apply learned skills into both performance and everyday situations Discussing how to maintain and deepen the mindfulness, acceptance and commitment skills after the program Mindfulness exercises Feedback from participants

**Table 2 ijerph-19-00134-t002:** Descriptive and inferential statistical overview of sociodemographic baseline characteristics between participants in the MAC and active control conditions.

	Study Condition					Statistics	
Variable	MAC		Active Control				
N	20		20				
	M	SD	M	SD	T (38)	*p*	d
Age (years)	22.00	2.30	21.45	2.52	0.59	0.56	0.18
Age range (years)	18–25		18–26				
Sport experience (years)	8.00	1.62	8.60	1.90	1.07	0.29	0.33
Weekly exercises	3.70	0.80	3.90	0.85	0.76	0.45	0.24
	*n*	%	*n*	%	χ^2^ (df = 1)	*p*	
Marital status					0.11	0.74	
Single	13	65	14	70			
Married	7	35	6	30			
Educational level					0.46	0.80	
High school	8	40	9	45			
Bachelor	8	40	6	30			
MSc	4	20	5	25			
Intensity of training					0.74	0.37	
Moderate	6	30	8	40			
Vigorous	14	70	12	60			
Team vs. individual sport							
Team	13	65	11	55			
Individual	7	35	9	45	0.42	0.52	

Notes: MSc = Master of Science; MAC = Mindfulness–Acceptance–Commitment.

**Table 3 ijerph-19-00134-t003:** Descriptive statistics for outcome variables at baseline, at post-intervention and at the end of the study, separately for participants in the Mindfulness–Acceptance–Commitment (MAC) and the active control condition.

Time Points						
N	Baseline		Post-Intervention		Study End	
	MAC	Active Control	MAC	Active Control	MAC	Active Control
	20	20	20	20	20	20
	M (SD)	M (SD)	M (SD)	M (SD)	M (SD)	M (SD)
Mindfulness	3.93 (0.53)	4.01 (0.55)	4.55 (0.51)	3.94 (0.54)	4.70 (0.50)	4.03 (0.58)
Self-compassion	3.52 (0.51)	3.41 (0.58)	4.01 (0.54)	3.56 (0.56)	4.14 (0.53)	3.55 (0.47)
Grit	3.54 (0.47)	3.53 (0.55)	3.92 (0.47)	3.51 (0.54)	4.06 (0.43)	3.61 (0.54)

Notes: M: mean, SD: standard deviation; MAC: Mindfulness–Acceptance–Commitment.

**Table 4 ijerph-19-00134-t004:** Inferential statistics for outcome variables at baseline, at post-intervention and at follow-up, separately for participants in the MAC = Mindfulness–Acceptance–Commitment (MAC) and the active control condition.

	Factors					
	Time		Group		Time × Group Interaction	
	F	η_p_^2^	F	η_p_^2^	F	η_p_^2^
Mindfulness	44.68 ***	0.54 (L)	6.06 *	0.13 (M)	46.80 ***	0.55 (L)
Self-compassion	37.15 ***	0.49 (L)	6.02 *	0.13 (M)	13.80 ***	0.27 (L)
Grit	15.50 ***	0.29 (L)	3.85	0.09 (M)	10.41 ***	0.21 (L)

Notes: Degrees of freedom: (1, 38); (M) = medium effect size; (L) = large effect size; * = *p* < 0.05; *** = *p* < 0.001.

**Table 5 ijerph-19-00134-t005:** Overview of effect sizes between group comparison at the end of the study and within group comparison from baseline to the end of the study.

Effect Size Comparisons (Cohen’s ds)			
	Between the MAC and Active Control Condition at the End of the Study	Within the MAC Condition from Baseline to the End of the Study	Within the Active Condition from Baseline to the End of the Study
Mindfulness	1.24 (L)	2.62 (L)	0.08 (T)
Self-compassion	1.19 (L)	2.07 (L)	0.37 (S)
Grit	0.92 (L)	1.45 (L)	0.17 (T)

Notes: Cohen’s ds: (T) = trivial effect size; (S) = small effect size; (L) = large effect size. No significant differences were found at baseline.

## Data Availability

Data are made available upon request from acknowledged experts in the field and upon a clear justification.

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
