# Peer review of "Examining the Effects of Mindfulness–Acceptance–Commitment Training on Self-Compassion and Grit among Elite Female Athletes"

_ijerph, 2021, doi:10.3390/ijerph19010134_

Round 1
Reviewer 1 Report
I consider this paper interesting and it can be an important addition to the literature. In particular, I appreciate the adopted design (a randomized trial with an active control group) and methodology (longitudinal study and psychometrically valid measurements). Although the limitations are described in the paper, the authors clearly state the research hypothesis and test them correctly (a generalized linear model approach would also ensure a better assessment of type II error, but it is not a mandatory procedure). The introduction is complete but the nomological network among constructs is non perfectly defined (but it reflects the contrasts present in the research literature). The study results are consistent with the adopted methodology, although the effect sizes are extremely high in comparison with the evidence from previous studies (also according to literature cited in the paper) and this fact should be taken into consideration. It should be also addressed, the strange trend (not carefully pointed out in my opinion) of the continuous raising of MAC group means over time (at study end) considering that from protocol doesn’t look like it was asked to the participants to continue mindfulness self-training.

Author Response
We thank Reviewer #1 for their valuable comments, which helped us to improve the quality of the manuscript. Please find the detailed point-by-point-response attached as a separate file.
Again, thank you very much for all your kind efforts.

Reviewer 2 Report
I congratulate the authors for their excellent work. It's an interesting and important study that is well written and consistent throughout the manuscript. The authors highlighted the importance of Mindfulness training, which could be an important key for performance.
The instruction is well structured. It contains all the necessary information to understand mindfulness and highlight the importance of these kinds of training. The methods part is perfect, and it has everything to understand the results. However, the result part needs some revision. The description of the result is superficial, and it feels like the readers can't decide what tables the authors introduce. Please restructure this part of the study. The conclusions are also well established, but it would increase the importance of the study if the authors put their results more into an international context.
Author Response
We thank Reviewer #2 for their valuable comments, which helped us to improve the quality of the manuscript. Please find the detailed point-by-point-response attached as a separate file.
Again, thank you very much for all your kind efforts.

Reviewer 3 Report
The study is interesting and in accordance with the subject matter of the journal. It is written in appropriate scientific language and the work is organized according to the sections of academic publications.
The abstract provides a clear understanding of the study that has been carried out. The title is aligned with the objective, the method used and the results obtained. The keywords are appropriate.
1. Introduction
The authors make an extensive review of the elements to be taken into account in the study, justifying the intervention they have carried out and the variables on which they will intervene.
The objective is well formulated.
The only change suggested by the reviewer is to eliminate the last paragraph of the section since its content seems more appropriate for the discussion section, as benefits of the work presented in the event that the hypothesized improvement is obtained.
2. Methods
The method section is correct. Well organized, well explained, the work done is perfectly understood.
3. Results
The section is satisfactory.
It is suggested to review small formatting mismatches in the tables. In general review when they put bold, italics, etc. to homogenize them. Specifically:
Table 2: line 8, titles, perhaps it should be separated from the top row so that the information is better understood or make a different table with that data.
Table 3: revise Mindfulness, as, at least in the revised version, there is an 's' out of place. Bold in the conditions compared in baseline, but not in the rest.
Table 4: in the text of the table footnote it is suggested to put *p<.05 together on the same line.
4. Discussion
It is suggested to make a more in-depth comparison with related studies to show the benefits and advantages obtained against the rest as well as the differential aspects of the literature consulted.
Author Response
We thank Reviewer #3 for their valuable comments, which helped us to improve the quality of the manuscript. Please find the detailed point-by-point-response attached as a separate file.
Again, thank you very much for all your kind efforts.

Reviewer 4 Report
First, I would like to thank the journal for giving me the opportunity to review such an interesting paper. I would also like to congratulate the researchers for the proposal they present. I find the subject matter very interesting, I am confident that training based on Mindfullness can provide tools of interest to athletes who are subjected to the demands of competition. As can be deduced from previous research that is cited in the introductory section. I also congratulate them for choosing female athletes as subjects of study, since they are a specific group, with their own needs and characteristics and there are many aspects to be investigated in the field of sport psychology.
In general, the work appears to me to be very correct in its development.The criteria for inclusion and exclusion of the participants are clear and the treatment of the data is very appropriate. In addition, the researchers include among their limitations some of the objections that could be proposed to them. For example, in lines 454 to 456 they state some reasons why they do not offer measures of whether the intervention improved objective athletes' performance.
There is just one issue that I believe is not adequately explained and requires more attention from the researchers. This is the intervention that the control group receives. From my point of view, it is explained in a very limited way. It gives the impression that its objectives have very little to do with the variables that are the object of evaluation in the study: Mindfulness, Grit and self-compassion.Since these dimensions are not usually among the typical objectives of interventions in sports psychology, this circumstance in its own right should be contemplated in some way among the limitations of the study. The variables evaluated are associated in a very specific way with Mindfulness-Acceptance-Commitment intervention In fact, the authors themselves recognize this in Lines 428-430 where they state that "the aim of Mindfulness training based on the MAC approach "is not to help the athletes to engage in the futile task of managing and controlling internal sport life. Rather, MAC enables athletes to increase their willingness to accept negative thoughts and emotions in pursuit of valued ends". I would like the authors to describe in greater detail the intervention developed by the control group and to what extent its objectives are related to the results obtained and, if necessary, to include it explicitly among the limitations of the study.
Author Response
We thank Reviewer #4 for their valuable comments, which helped us to improve the quality of the manuscript. Please find the detailed point-by-point-response attached as a separate file.
Again, thank you very much for all your kind efforts.
